# Assessment of Dynamic Bayesian Models for Gas Turbine Diagnostics, Part 2: Discrimination of Gradual Degradation and Rapid Faults

**Valentina Zaccaria \*, Amare Desalegn Fentaye and Konstantinos Kyprianidis** 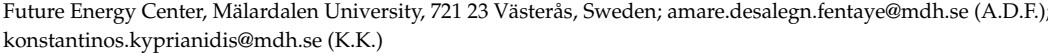

Future Energy Center, Mälardalen University, 721 23 Västerås, Sweden; amare.desalegn.fentaye@mdh.se (A.D.F.); konstantinos.kyprianidis@mdh.se (K.K.)
\* Correspondence: valentina.zaccaria@mdh.se

**Abstract:** There are many challenges that an effective diagnostic system must overcome for successful fault diagnosis in gas turbines. Among others, it has to be robust to engine-to-engine variations in the fleet, it has to discriminate between gradual deterioration and abrupt faults, and it has to identify sensor faults correctly and be robust in case of such faults. To combine their benefits and overcome their limitations, two diagnostic methods were integrated in this work to form a multi-layer system. An adaptive performance model was used to track gradual deterioration and detect rapid or abrupt anomalies, while a series of static and dynamic Bayesian networks were integrated to identify component degradation, component abrupt faults, and sensor faults. The proposed approach was tested on synthetic data and field data from a single-shaft gas turbine of 50 MW class. The results showed that the approach could give acceptable accuracy in the isolation and identification of multiple faults, with 99% detection and isolation accuracy and 1% maximum error in the identified fault magnitude. The approach was also proven robust to sensor faults, by replacing the faulty signal with an estimated value that had only 3% error compared to the real measurement.

**Keywords:** gas turbine diagnostics; Bayesian network; hybrid models

## 1. Introduction

The intense competition in the energy sector has been driving the progress in assets health management systems with the aim of reducing maintenance and operating costs [1]. An effective gas turbine diagnostics system requires comprehensive monitoring of both gradual deterioration mechanisms and possible malfunctions in any of the system components and ancillaries. Since the aim of diagnostics is to provide information to take the best possible maintenance action, discriminating between gradual degradation and abrupt faults, which may occur at the same time, is essential. This problem has been addressed in the literature mostly by integrating different diagnostic techniques for these two types of faults [2,3]. For example, tracking filters were used to monitor gradual deterioration and update the engine baseline, while other data-driven techniques were used for online anomaly detection [4]. In all cases, the diagnostics system needs to be robust enough to detect and identify sudden faults in presence of deteriorating components [5,6]. In [7], a combination of model-based gas path analysis (GPA) and auto associative neural network was proven effective in discriminating among component deterioration, rapid faults in the rotating components, and bleed valve faults.

The major distinction between gradual and abrupt faults in a gas turbine system lies in the time over which the fault forms and evolves [3]. Gradual degradation usually affects more than one component at a time, it is caused by expected long-term phenomena such as mechanical wear or gradual particles deposition, and it typically manifests as gradual measurements deviations over time from the design values. On the contrary, an abrupt fault can be seen as a sudden performance loss due to a single-time event such as foreign

object ingestion, a valve stuck in a wrong position, a sensor malfunction, etc. Statistically, abrupt faults are rare and can be therefore assumed to occur for a single component at a time [8]. In the literature, the terms abrupt fault and rapid fault have been distinguished depending on whether the fault magnitude stays constant or keeps increasing [8]; however, in this article, the two terms are used interchangeably. In the present work, examples of gradual degradation are compressor fouling (CF) and turbine erosion (TE), while examples of abrupt faults are bleed valve leakage (BV) and sensor faults (bias).

Tracking of gradual deterioration has been commonly achieved with Kalman filter and its derivative, or with the aid of physics-based performance models [9–11]. A drawback for these methods is that they usually require a large number of measurements, and their accuracy is limited when considering a practical sensor set [12]. The significant variations among machines of the same fleet is another limitation for the accuracy of model-based diagnostics [13]. In the past decade, a remarkable increase in artificial intelligence techniques has been observed for both trend monitoring and abrupt faults identification [14–16]. Despite the very positive results obtained by such diagnostic methods, a limitation lies in the wide amount of data needed for training purposes. In fact, the quality of the diagnostics system is only as good as the underlying data [7]. This presents a non-trivial challenge, due to the underrepresentation of faulty samples in historical data, especially when it comes to simultaneous multiple faults [13,17].

Combinations of physics-based models and data-driven methods were proposed as a solution to overcome several shortcomings of the single methods [18–21]. In particular, integration of an adaptive performance model with Bayesian networks (BNs) or with artificial neural networks (ANNs) was proposed by the authors of this paper [7,22]. The challenge with multiple fault scenarios in BNs is that the size of the conditional probability tables (CPTs—the numerical relationships between events) increases exponentially [23]. To keep the problem tractable, hierarchical BN models were proposed [24,25]. However, this solution was not applied to the challenge of discriminating between gradual degradation and abrupt faults.

Since both physics-based and data-driven methods for diagnostics rely on correct measurement information, it is essential to isolate sensor malfunctions, which may lead to an erroneous fault signature [26]. Sensors, as any other component, tend to degrade over time and are subject to anomalies such as bias, erratic signal, or stuck reading [26,27]. Especially in presence of an excessive bias, the resulting measurements signature may be confused with a process fault. Several approaches for sensor fault detection have been proposed, and there is common agreement that sensors monitoring should be a step in the diagnostics routine [27–29]. Any bias or wrong reading should be detected before proceeding with the identification of process faults. Gaussian reconciliation technique and wavelet analysis are two examples of common approaches that detect anomalies in the signal distribution [26,27]. With the same principle, a BN can be used to relate sensor data distribution to the probability of a fault. In addition, the BN for sensor fault detection can be integrated into the hierarchical BN system proposed in this work.

For the first time in this work, four BNs were integrated with a steady-state performance model for the purpose of isolating and identifying gradual components degradation and abrupt faults. The design and assessment of the BNs was presented in Part 1 of this paper, which provided a comparison between static and dynamic BNs [30]. The application on synthetic and field data and the robustness against sensor faults are evaluated in this Part 2. The novelty of the work is the integration of physics-based models with hierarchical BNs for discrimination of gradual degradation and abrupt faults occurring simultaneously, which addresses the following challenges: (i) correct identification of fault magnitude in presence of underlying degradation, (ii) modularity of BN models and tractability of the CPTs when considering multiple faults, (iii) reduced dataset required for training the BN models since they can be trained with single-fault scenarios, and (iv) robustness in case of sensor faults.

## 2. Methods

From a diagnostics perspective, differentiating between gradual degradation and abrupt faults is possible by comparing the measurements at the current time t with both the values at the time t − 1 and the reference values in healthy conditions. When only gradual degradation occurs, the difference in performance between time t and t − 1 is very close to zero, if a sufficiently small time step is chosen. The difference from the reference condition indicates the degradation severity and triggers an alarm if this surpasses a safe threshold. On the other hand, an abrupt fault can happen at any time, either on a fully healthy machine or on a degraded one. The unique signature of an abrupt fault manifests as a notable difference in performance between two consecutive time steps. Comparing the performance with the reference condition is not a good indicator of the fault severity, because this difference is partly due to the continuous degradation the machine was undergoing until this time [6].

Compressor fouling is one of the most common causes of degradation in a gas turbine and it was therefore included in this work [31]. Other common degradation phenomena in industrial gas turbines are tip clearances increase in both compressor and turbine, and trailing edge erosion in turbine blades [31]. For the demonstration of the proposed hierarchical models, only compressor fouling and turbine erosion were considered in this work to simulate gradual degradation. The inclusion of other phenomena like clearance increase could be possible by extending the training space of the BNs, although discrimination between different degradation phenomena is beyond the scope of this work. Other typical faults of interest are bearing faults, which have been extensively covered in the literature [32,33], faults in the fuel injection system [34], or bleed valve fault, which instead has been more neglected in the literature. Bleed valve leakage is a particularly interest case for diagnostics because its measurements signature can be very similar to the one of turbine degradation [35].

### 2.1. Gas Turbine Model

To diagnose multiple simultaneous faults, a multi-layer system is proposed in this work. The first layer is composed of a physics-based, non-dimensional model, which was already introduced in Part 1 of this work [30]. The performance model of the gas turbine system was extensively described in previous work [22,35,36]. Compared to [22], and similarly to [35], the model was modified to simulate a single-shaft gas turbine connected to a generator. The gas turbine under consideration is the Siemens Energy SGT800 (Munich, Germany), a single-shaft 50 MW class turbine. The validity of the model was proven against reference data in [35]. For diagnostics purpose, the model computes a Jacobian matrix between health state variables (deviations in efficiency and flow capacity) and measurement residuals, and it uses the Broyden method to estimate the values of the deviation factors that minimize the residuals.

A variation of the scheme applied in [22] is proposed here. The deviation factors considered in this work are for the efficiency and flow capacity for both the compressor and the turbine. The measurements used for the matching scheme and the Jacobian matrix are as follows: the compressor outlet temperature (T3) and pressure (P3), the compressor inlet flow rate (W2), which is measured through a bellmouth intake, and the turbine exhaust temperature (T5). A method similar to the one proposed in [12] was used to determine the optimal measurements for the scheme. The schematic of the gas turbine system and the sensors' location is presented in Figure 1.

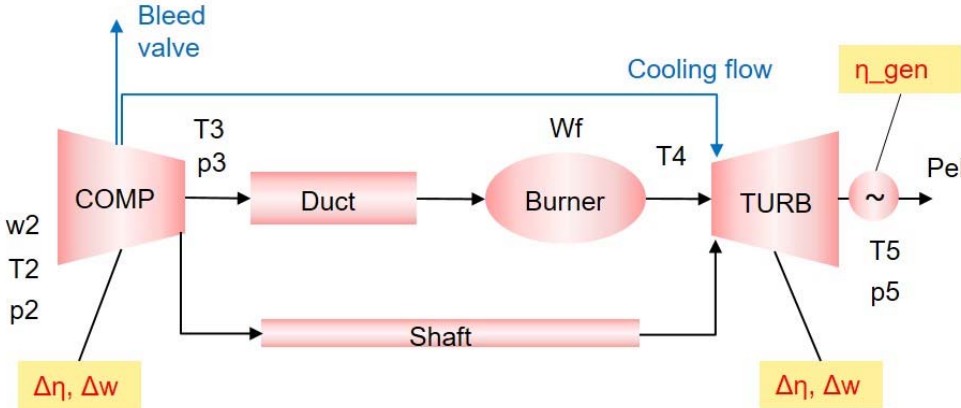

**Figure 1.** Schematic diagram of the gas turbine model ($\eta$_gen: generator efficiency).

It is worth noting that W2 is not always measured in gas turbines, requiring a modification of the matching scheme proposed here. Given the high correlation between W2 and the compressor flow capacity change, this deviation may be more difficult to identify correctly with a reduced set of sensors. For two-shaft machines, the rotational speed of the high-pressure turbine could be used instead.

When the model runs in diagnostics mode, the inputs are the measurements from the real plant, the power, the ambient conditions (temperature, pressure, and humidity), and the inlet guide vane (IGV) position. The deviation factors in efficiency and flow capacity are the outputs. The deviation factors were calculated as per Equations (1) and (2); hence, deviation in efficiency is zero in healthy condition while the flow capacity deviation factor is 1.

$$\Delta \eta = \eta_{fault} - \eta_{healthy} \tag{1}$$

$$\Delta \overline{w} = \frac{\overline{w}_{fault}}{\overline{w}_{healthy}} \tag{2}$$

Once the deviation factors have been estimated, these values are used as new inputs to run the model to a set of reference conditions. For this step, all the other inputs (ambient conditions and power) are kept at reference conditions. The new outputs are the measurements residuals, calculated according to Equation (3).

$$r = \frac{z - z_{ref}}{z_{ref}} \tag{3}$$

Residuals are calculated for T3, P3, T5, P5 (turbine exhaust pressure), W2, and Wf (fuel flow). Reference conditions in this work refer to the conditions presented in Table 1.

**Table 1.** Reference conditions for the gas turbine model.

| | |
|---|---|
| Ambient temperature ($T_{amb}$) | 298.15 K |
| Ambient pressure ($p_{amb}$) | 101.325 kPa |
| Relative humidity (RH) | 60% |
| GT power | 50 MW |

*2.2. Muti-Layer Diagnostic System*

With the purpose of identifying different fault scenarios, the model is integrated with a series of Bayesian networks (BNs), which receive the measurement residuals at reference conditions as inputs. Four different BNs were developed to differentiate between gradual compressor degradation, gradual turbine degradation, an abrupt fault (bleed valve leakage as an example), and sensor faults. The steps performed in the diagnostic system are illustrated in Figure 2.

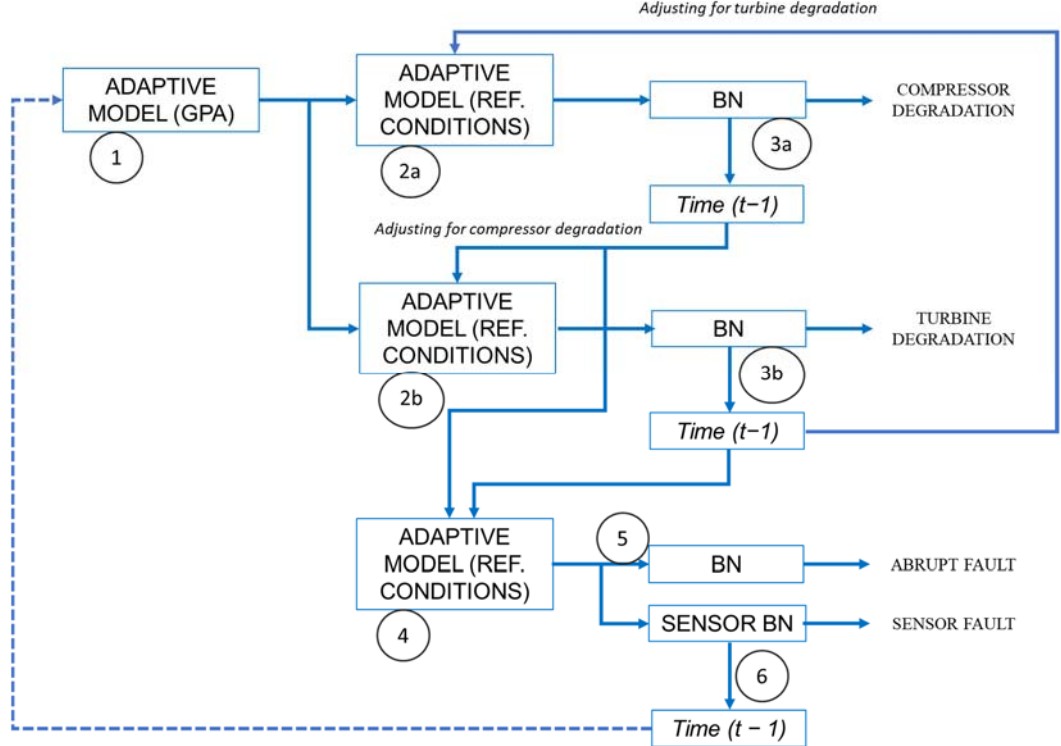

**Figure 2.** Schematic diagram of the multi-layer diagnostic system based on adaptive performance model and Bayesian networks.

1     Estimate performance deviation factors in the rotating components via means of gas path analysis (GPA) with the performance model;

2a    Adjust for turbine degradation: take the difference between turbine performance factors at time t and t − 1 to use in the next step, then simulate the GT system at reference (REF) conditions using the compressor performance factors from step 1 and the adjusted turbine performance factors;

2b    Adjust for compressor degradation: take the difference between compressor performance factors at time t and t − 1 to use in the next step, then simulate the GT system at reference conditions using the turbine performance factors from step 1 and the adjusted compressor performance factors;

3a    Feed the results of step 2a to a dynamic Bayesian network to identify compressor degradation, which gives the compressor performance for time t + 1;

3b    Feed the results of step 2b to a dynamic Bayesian network to identify turbine degradation, which gives the compressor performance for time t + 1;

4     Adjust for compressor and turbine degradation: performance factors for both compressor and turbine are taken as the difference between t and t − 1 and the model is run at reference conditions to isolate the effect of rapid or abrupt faults;

5     Feed the residuals from step 4 to a BN to identify an abrupt fault (e.g., BV leakage);

6     Feed the residuals from step 4 to the final BN to identify a sensor fault, and send the information to adjust the matching scheme at the next time step t + 1.

The purpose of steps 2a, 2b, and 4 is firstly to remove deviations caused by off-design ambient and operating conditions, as in [22]. Secondly, the performance is also adjusted to remove previously identified deviation in turbine performance (step 2a) or compressor performance (step 2b) or both (step 4) in order to separate degradation assessment from abrupt fault identification. In this way, separate BNs can be utilized to assess simultaneous degradation and faults in different components, avoiding extremely large CPTs. When a sensor fault is detected, the GPA scheme in step 1 is modified according to the method proposed in [37], since the values from the faulty sensor do not provide reliable information to estimate the performance deviations.

### 2.3. Bayesian Networks

Dynamic Bayesian networks (DBN) for CF and TE were developed as discussed in Part 1 of this work [30]. The parent nodes in these models represented the degradation severity for CF and TE, for the respective network, while the measurement residuals constituted the child nodes. The degradation severity is defined as reported in Equation (4).

$$S = -\Delta\eta \cdot \sqrt{1 + \left(\frac{\Delta\overline{w} - 1}{\Delta\eta}\right)^2} \tag{4}$$

A fault severity between 0 and 0.5% was considered Normal (N), between 0.5 and 1% Very Low (VL), between 1 and 2% Low (L), between 2 and 3% Medium (M), and above 3% High (H). The parent nodes were created as temporal nodes, i.e., the prior probability was set as a function of the previous condition, which was proven to achieve higher diagnostics accuracy than considering a constant prior probability distribution [30]. Each network was trained with data generated by the performance model described in Section 2.2, by simulating data from various machines. Deviations between machines in the same fleet were taken into account with different design values for efficiency and flow capacity, different ratios $\frac{\Delta\overline{w}-1}{\Delta\eta}$ between 1 and 2, and measurement noise. The training phase was used to estimate the CPTs for each node by using the Maximum Likelihood Estimator [38].

Static BNs were instead employed for BV leakage and sensor faults, where the single event probability is independent from the previous condition. The structure of the BV BN was similar to the one for CF and TE, with the parent node being the BV fault represented by 5 states or fault severities. The fault severity in this case represented the leakage flow in percentage of the turbine flow, where 0% is N, between 0 and 1% VL, between 1 and 2% L, between 2 and 3% M, and above 3% is H. BV leakage is not a gradual phenomenon, it can occur suddenly with any severity. The initial probability was set according to Table 2.

**Table 2.** Prior probability distribution $P(Y)$ for BV leakage.

| N | VL | L | M | H |
|---|---|---|---|---|
| 99% | 0.25% | 0.25% | 0.25% | 0.25% |

A fourth BN for detecting sensors faults is depicted in Figure 3. Each child node was composed of five states: normal (N), low reading (L), very low (VL), high reading (H), and very high (VH). The parent nodes can assume the states normal (N) or faulty (F), the latter indicating a sensor fault. Note that only the sensors that are used for the diagnostic scheme were included in this work, but the principle can be extended to multiple sensors. The definition of low and high reading is the measurement residual being between −0.5% and −1.5%, and between 0.5% and 1.5%, respectively. Very low and very high indicate residuals below −1.5% and above 1.5%, respectively.

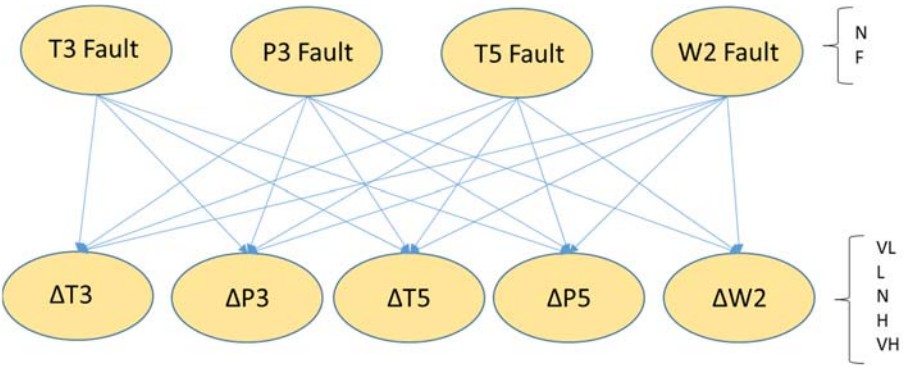

**Figure 3.** The static BN for sensor faults detection (sensor state is normal (N) or faulty (F)).

### 2.4. Tested Scenarios

Three scenarios were tested to assess the capability of the proposed diagnostics approach. First, synthetic data were generated with the performance model to simulate simultaneous compressor fouling, turbine erosion, and bleed valve leakage. The gradual degradation in compressor and turbine components was simulated as a linear decay in efficiency and flow capacity, with a ratio $\frac{\Delta \overline{w} - 1}{\Delta \eta}$ equal to 1.5. In particular, both compressor efficiency and flow capacity were decreased to simulate fouling, while turbine erosion was simulated by an increase in turbine flow capacity and a decrease in efficiency.

A total of 6000 h (1 data point per hour) were simulated, varying the power, the ambient temperature, and the IGV position according to daily schedules. Note that the choice of degradation level was arbitrary, only for test purposes, and does not represent actual degradation of the SGT800 or any real engine.

In Scenario 2, field data from a Siemens Energy SGT800 machine were employed to validate the approach. Data from a month of operations were analyzed, containing an occurrence of BV leakage at day 21. Over this month, the engine was assumed to be in a state of no or very low degradation, although the exact conditions were unknown at the time of the test. In Scenario 3, the same data were also corrupted with simulated sensors faults. These scenarios are shown in Table 3.

**Table 3.** Test scenarios.

| Scenario | Gradual Compressor Degradation | Gradual Turbine Degradation | Abrupt Fault |
|---|---|---|---|
| 1: Simulated | From 0% to 4% Maintenance at 2500 h | From 0% to 3% | BV 2% |
| 2: Field data | Unknown | Unknown | BV unknown% |
| 3: Field data + simulated | Unknown | Unknown | T3 and P3 faults 12.5% |

All the synthetic and field data were first used as inputs to the performance model in diagnostic mode, and deviation factors were estimated. The sequence of steps 1–6 from the previous section was followed.

## 3. Results and Discussion

### 3.1. Scenario 1

The daily variations of power set-point and ambient temperature are depicted in Figure 4. These are used as inputs for the model together with the sensor outputs.

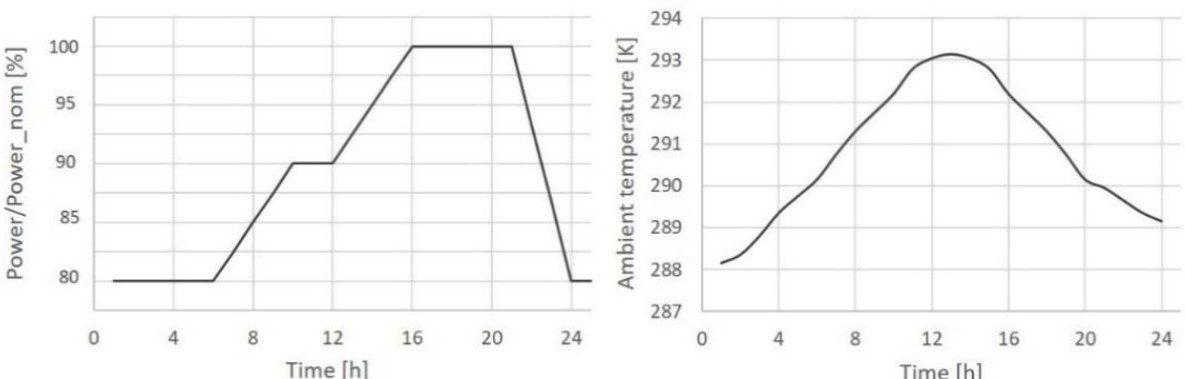

**Figure 4.** Daily power (**left**) and ambient temperature (**right**) profiles.

The results from the first simulated scenario are presented in Figures 5–7, which show the compressor and turbine degradation estimated by the adaptive model (Figures 5 and 6) and by the DBNs (Figure 7). The trends in efficiency and flow capacity deviations for both rotating components are clearly followed, as seen in Figures 5 and 6. The BV leakage events

were interpreted by the model's adaptation scheme as deviations in turbine efficiency and flow capacity, however, these erroneous deviations were not picked up by the DBN (Figure 7). It is also possible to note that the effect of operating power and ambient temperature is taken into account in the model, and does not hinder the correct estimation of the degradation conditions. Only some errors occurred in the CF prediction from the DBN around 2500 and 5000 h, probably due to excessive noise in the data, as seen at the top of Figure 7.

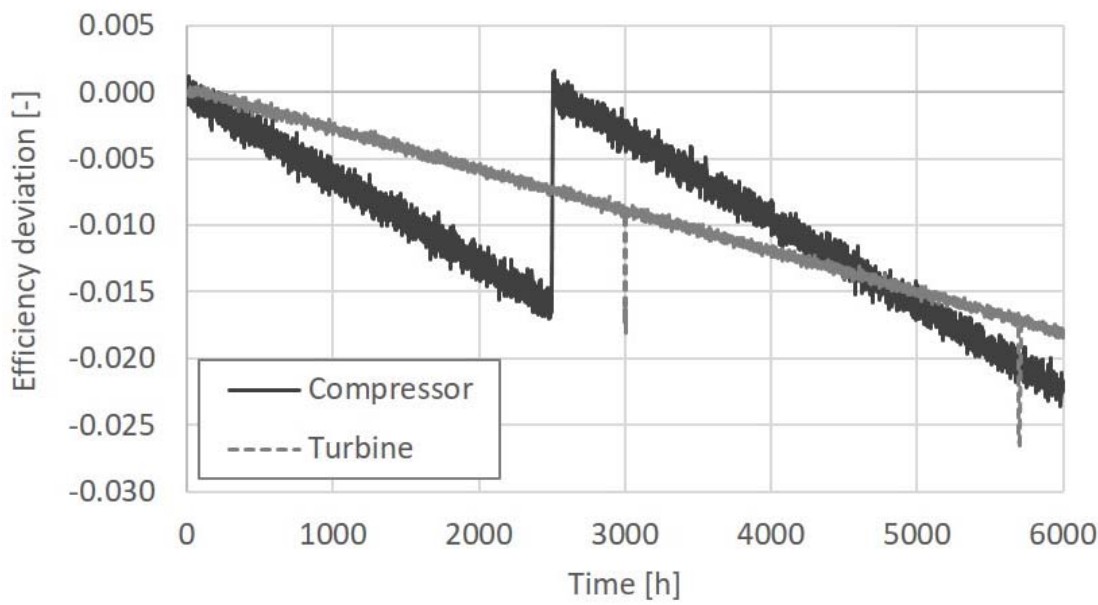

**Figure 5.** Estimated compressor and turbine efficiency deviations from the performance model in Scenario 1.

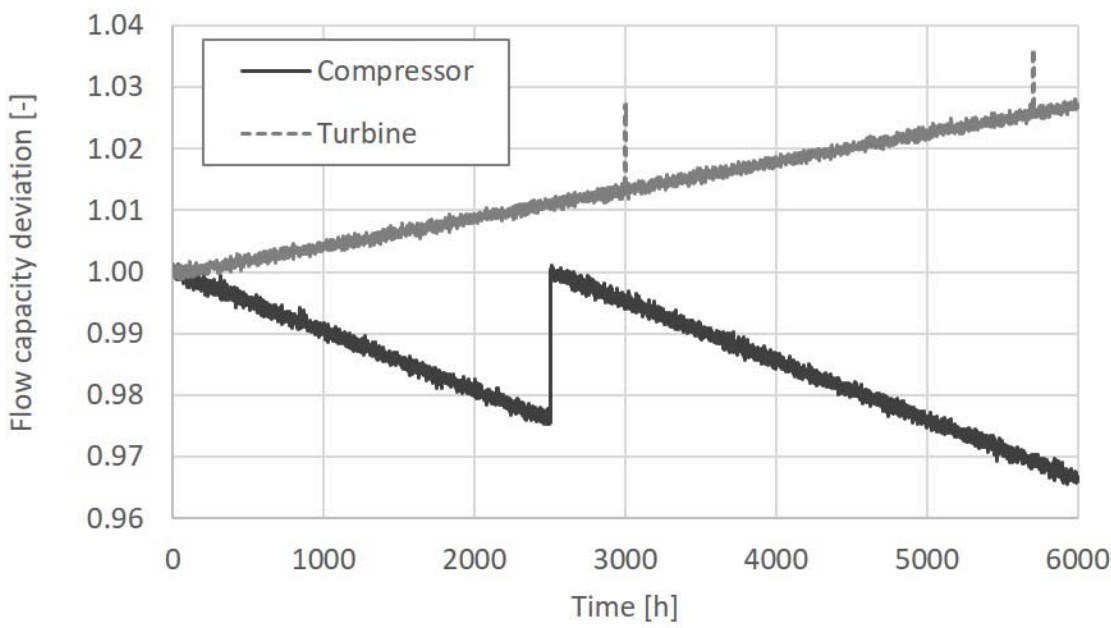

**Figure 6.** Estimated compressor and turbine flow capacity deviations from the performance model in Scenario 1.

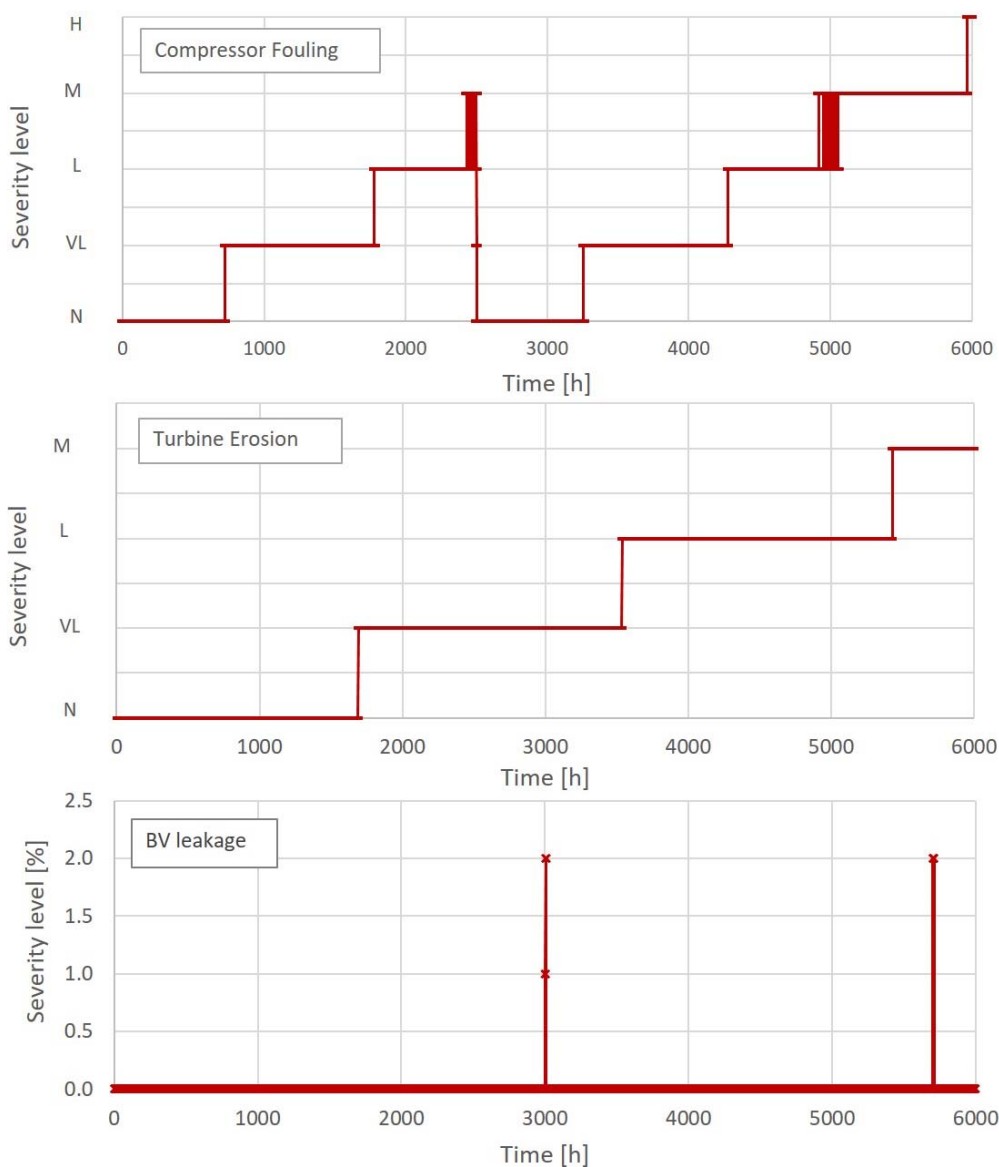

**Figure 7.** BN predictions of compressor fouling, turbine erosion, and BV leakage in Scenario 1. CF and TE are quantified from normal (N—0% degradation) to high (H—4% degradation).

### 3.2. Scenario 2

The data for Scenario 2 were taken from a Siemens Energy SGT800 machine operating over a month. On day 21, a BV leakage was reported. An extraction of the data is presented in Figure 8, where the performance factors calculated by the adaptive model are shown for compressor and turbine for day 1, day 10, and day 20. The y-axes show the deviations in efficiency and flow capacity as defined in Equations (1) and (2). It can be noted that compressor efficiency and flow capacity deviations oscillate a bit during this time, bounded between 0 and −1.5% for the efficiency and between 100% and 98% for the flow capacity. Turbine performance deviations are instead more stable and do not show any significant fluctuation over this period of time. This represents the baseline against which the analysis for day 21 was performed.

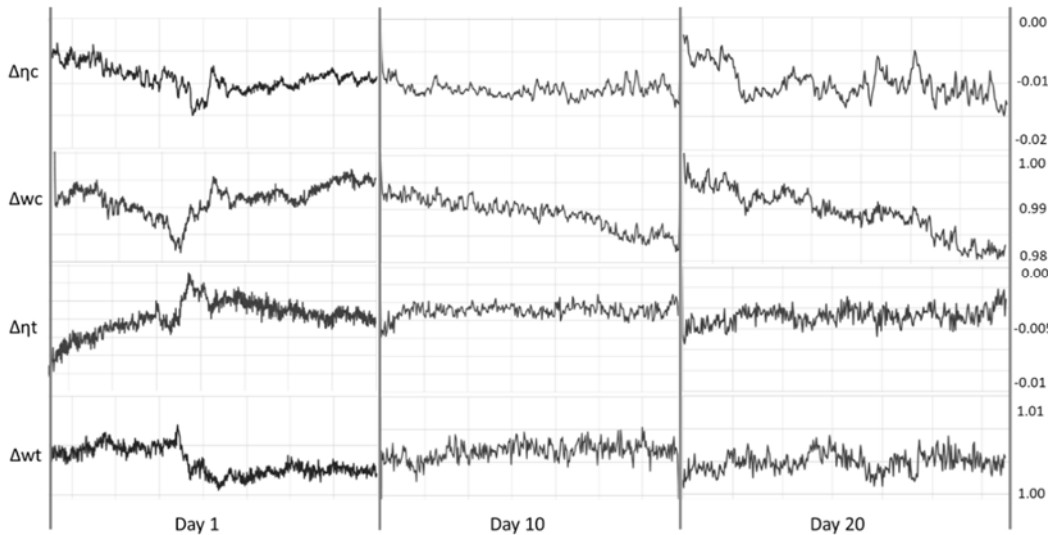

**Figure 8.** Estimated compressor efficiency ($\Delta\eta_c$), compressor flow capacity ($\Delta\overline{w}_c$), turbine efficiency ($\Delta\eta_t$), and turbine flow capacity ($\Delta\overline{w}_t$) deviations in the first 20 days of Scenario 2.

The estimated performance deviation factors for day 21 (i.e., during BV leakage) are shown in Figures 9 and 10, where the numerical values are removed for confidentiality reasons.

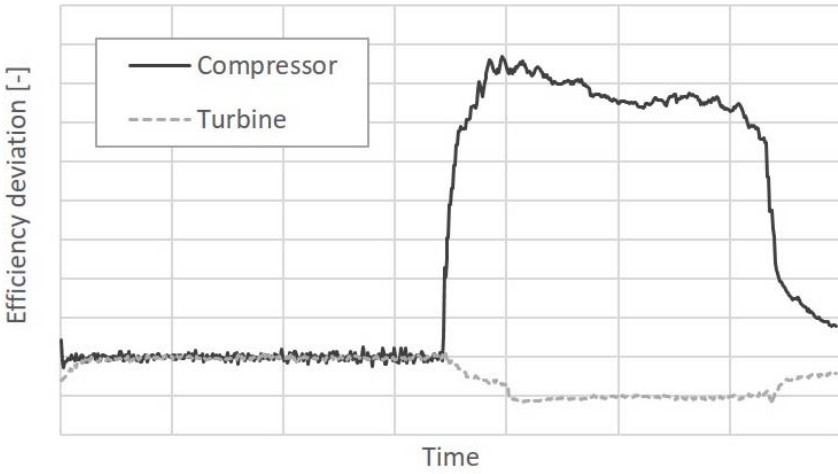

**Figure 9.** Estimated compressor and turbine efficiency deviations in day 21 of Scenario 2.

As Figures 9 and 10 show, the model adaptation to the measured variables resulted in an increment in estimated compressor efficiency and a simultaneous reduction in turbine efficiency and flow capacity. This efficiency increment was not physical, but a result of the model adaptation to the measurements in presence of BV leakage, in particular the varying ratio T3/P3 due to a change in operating point. The estimated turbine flow capacity decreased and then increased again as a result of the model matching scheme. It is possible to conclude that an abrupt fault occurred, because the changes in Figures 9 and 10 are very sharp. However, the results from the GPA cannot directly point to the fault type. The BN layers were in any case able to discriminate between gradual degradation and abrupt fault, as it can be seen in Figure 11. The two DBNs for gradual degradation did not react to the sudden change in performance since the probability to jump to a much higher degradation level was constrained to be very small. The third BN instead identified a BV leakage (bottom of Figure 11). In the last part, the BN predicted a reduction in the leakage, consistently with the trends in Figures 9 and 10. This was probably due to the fact that the power was decreased, before shutting down the machine and inspecting the valve.

Compressor degradation was also detected in the last minutes before shut-down, probably erroneously due to a quick load transient.

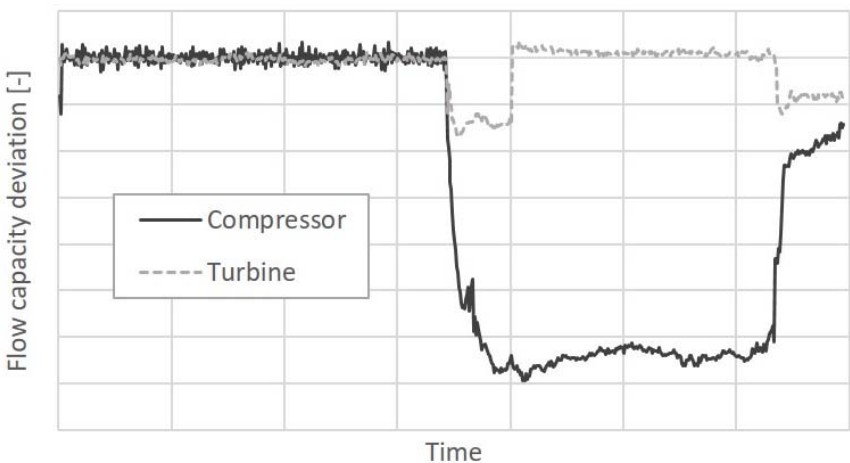

**Figure 10.** Estimated compressor and turbine flow capacity deviations in day 21 of Scenario 2.

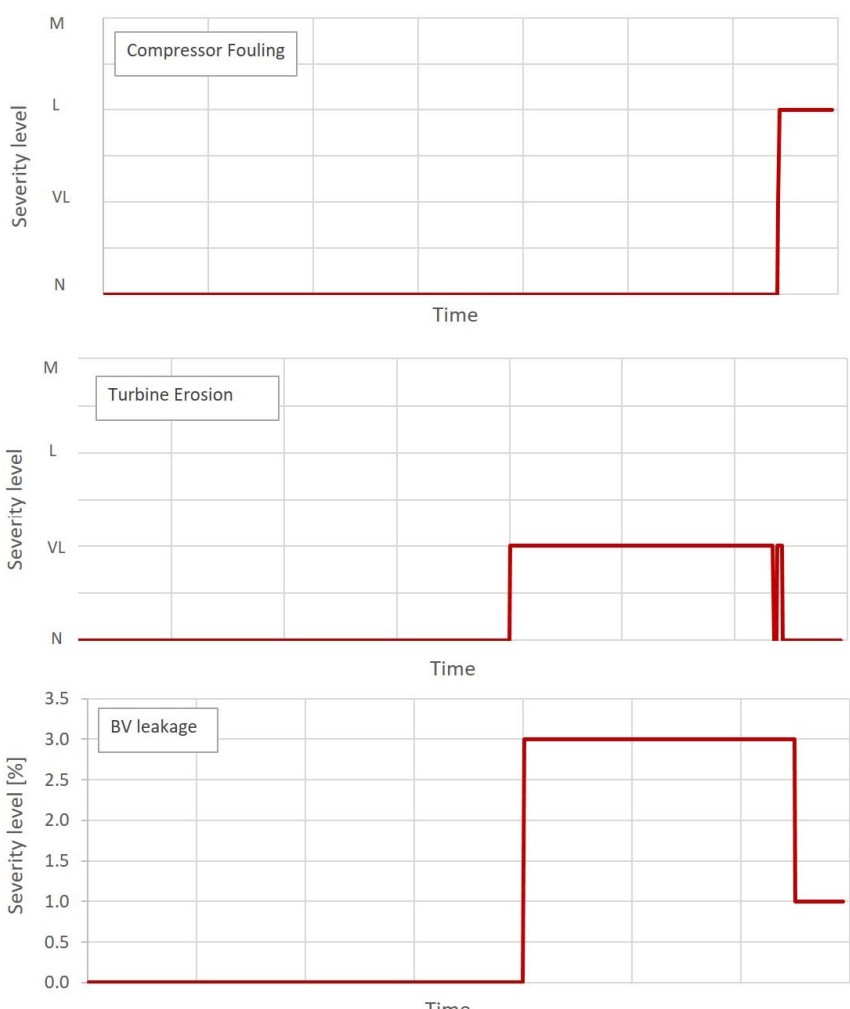

**Figure 11.** BN predictions of compressor fouling, turbine erosion, and BV leakage in Scenario 2.

### 3.3. Scenario 3

For the last Scenario, healthy SGT800 measurement data were corrupted by adding manually a sensor fault. The fault was simulated in this way by subtracting a fixed value from the real measurement, to simulate a sudden bias equal to 12.5% of nominal T3 and P3 values. The residuals of the measured variables after the manual fault injection are presented in Figures 12 and 13 for T3 and P3 fault, respectively.

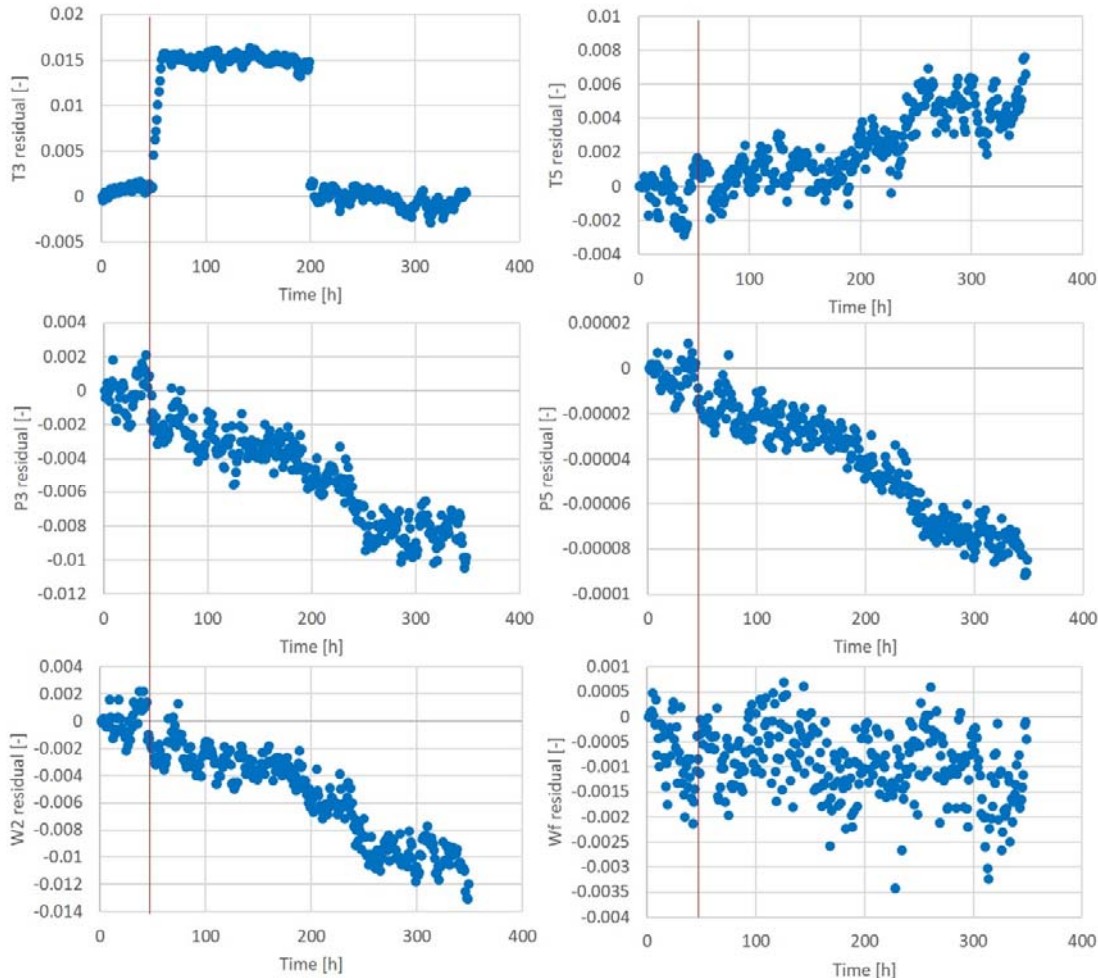

**Figure 12.** Measurement residuals after injected fault in T3 sensor, *y*-axis is the absolute residual (the fault was implemented at time 50, as indicated by the red line).

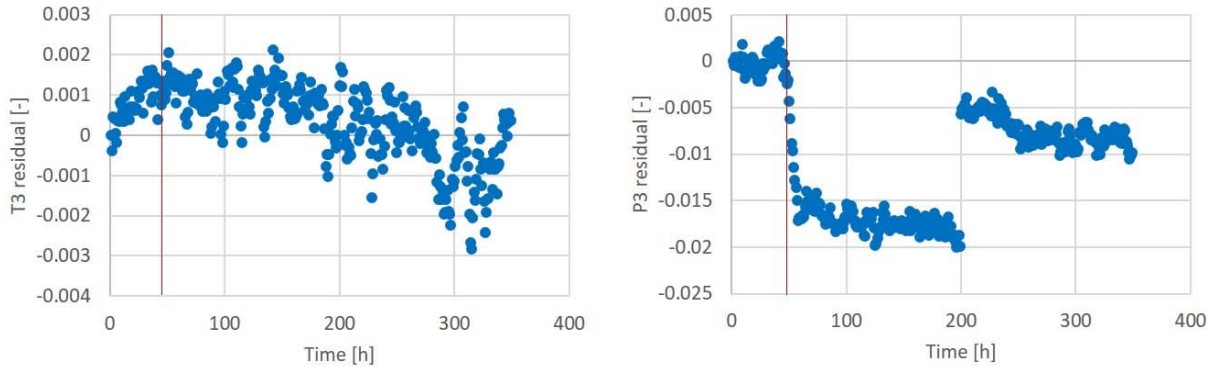

**Figure 13.** T3 and P3 residuals after injected fault in P3 sensor (the other residuals are identical to Figure 12).

As expected, the BNs for compressor degradation, turbine degradation, and BV leakage did not detect any anomaly. The BN for sensor faults instead detected the correct faults after the injection time, as illustrated in Figures 14 and 15.

According to the scheme presented in Figure 2, the first diagnostics layer requires four functioning sensors for the adaptive model to estimate the performance deviation factors. If one of the four sensors (T3, P3, T5, and W2) is faulty, the step 1 in the diagnostics scheme cannot proceed, because a square Jacobian matrix cannot be built for the analysis. Therefore, the method suggested in [37] was implemented to remove one performance deviation factor from the matching scheme and perform the GPA with a $3 \times 3$ matrix. The method from [37] established that $\Delta\eta_c$ (compressor efficiency) should be removed in case of T3 sensor fault, while $\Delta\eta_t$ (turbine efficiency) was to be dropped in case of P3 sensor fault. The previous values of $\Delta\eta_c$ and $\Delta\eta_t$ were instead used as constant inputs, respectively. T3 and P3 were then calculated in step 2 as normally, with the three current estimated deviation factors from step 1 and with the constant factor from the previous time step. The calculated values are compared with the real ones in Figures 16 and 17.

Even with a sensor missing from the matching scheme, the model was still able to produce suitable residuals for the underneath BNs. As depicted in Figure 16, the error between calculated and real T3 was below 2 K (less than 3%), while the one for P3 was below 15 kPa (or less than 7%). In conclusion, the BNs could still provide information about the status of the machine while the faulty sensor gets to be replaced.

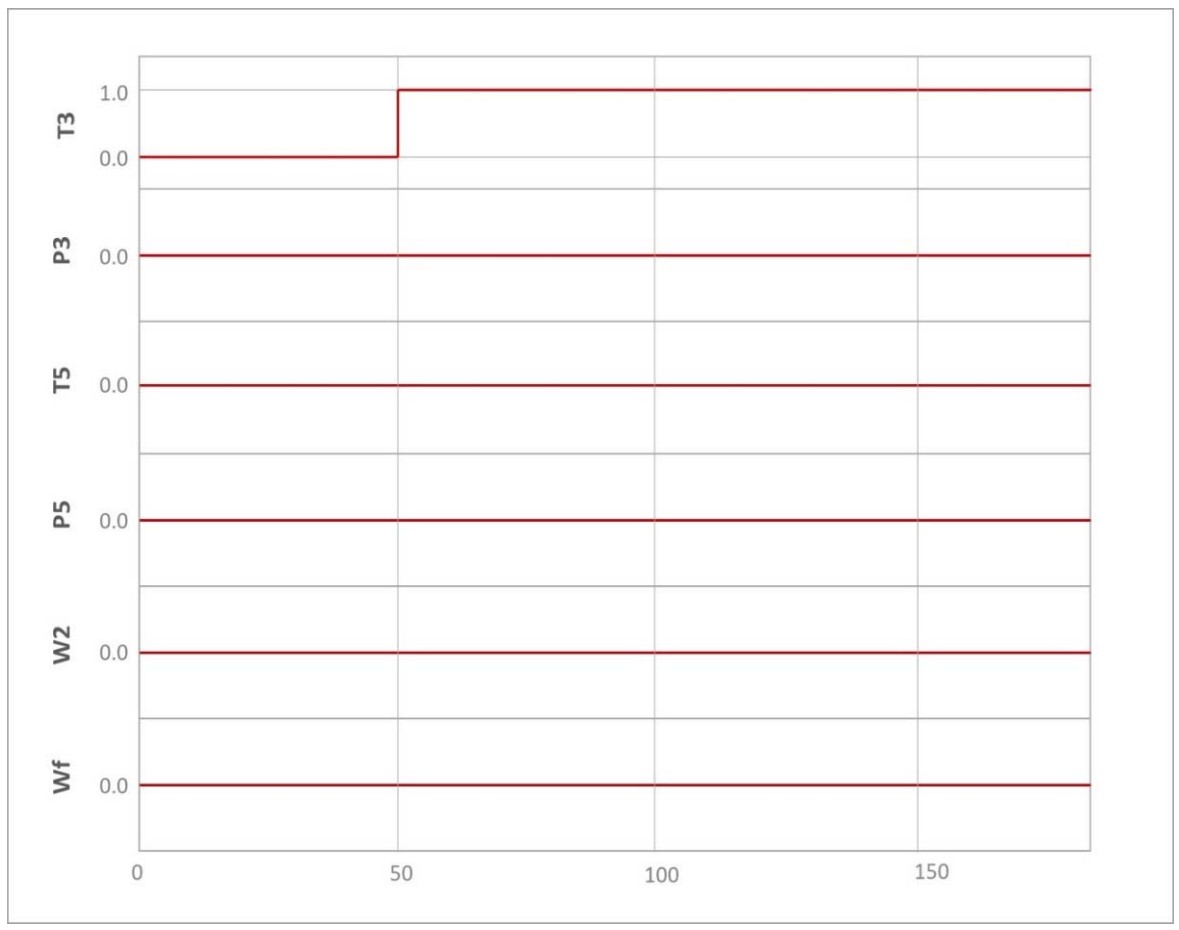

**Figure 14.** BN predictions of sensor status in Scenario 3 for T3 sensor fault. 0 means no fault and 1 means fault.

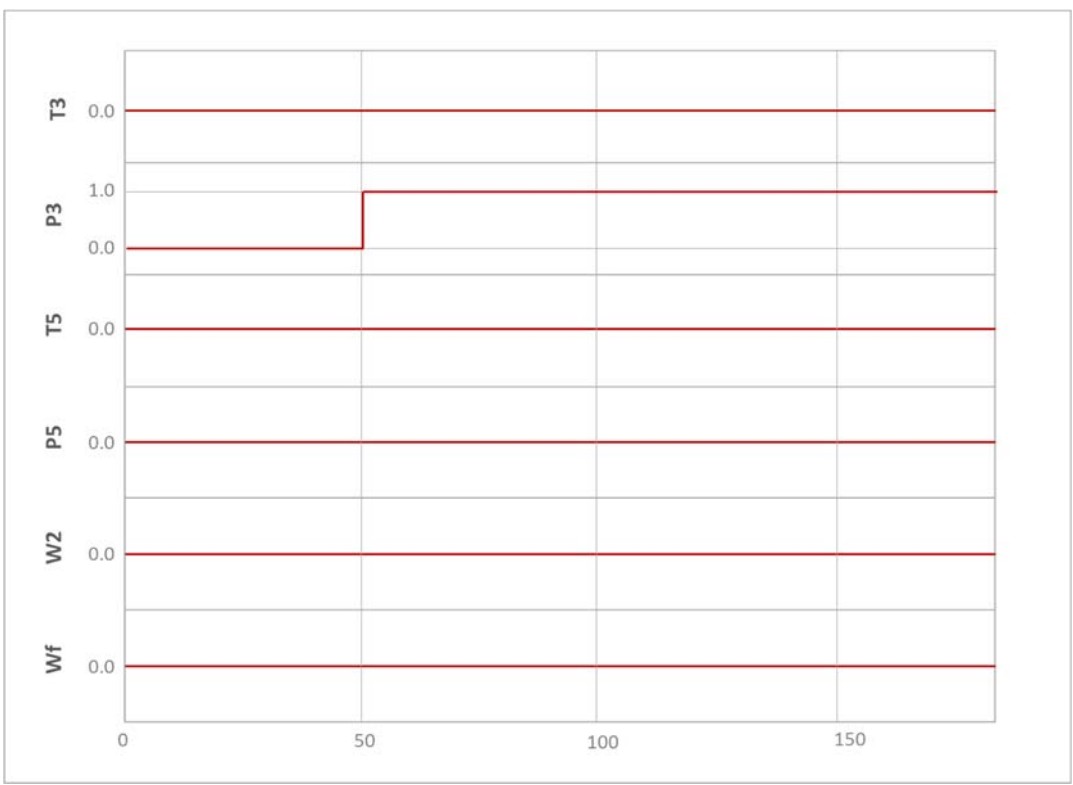

**Figure 15.** BN predictions of sensor status in Scenario 3 for P3 sensor fault. 0 means no fault and 1 means fault.

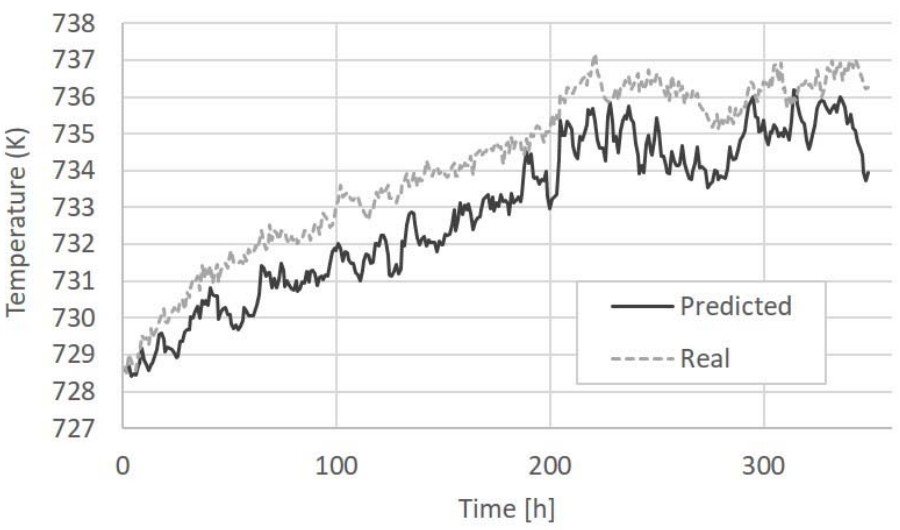

**Figure 16.** Predicted value of T3 vs. actual value.

### 3.4. Adaptability to Different Machines

In the previous Scenarios, the diagnostics system was applied to a single machine at a time, either a simulated one or a real one. Therefore, its application was demonstrated only on two gas turbines of the same class. In this Section, results are presented for the validation of the method on 2000 machines. The different engines were simulated with the gas turbine model, by performing Monte Carlo sampling on Gaussian distributions for efficiency and flow capacity of each rotating component, with $3\sigma$ equal to 0.5%. Since different machines degrade at different rates, the ratio $\frac{\Delta \overline{w}-1}{\Delta \eta}$ was varied according to a uniform distribution between 1 and 2. The same fault types as in Scenarios 1 were tested for each of the 2000 machines. The results of the method accuracy are summarized in Table 4,

where it is possible to see how the proposed approach can be applied to different units and maintain the desired accuracy. In Table 4, the true positive rate refers to the data points correctly classified as faulty, the false positive rate (or false alarms) represents healthy data points erroneously classified as faulty, while the true negative rate refers to the healthy data point correctly classified and the false negative rate to the faulty data point erroneously classified as healthy.

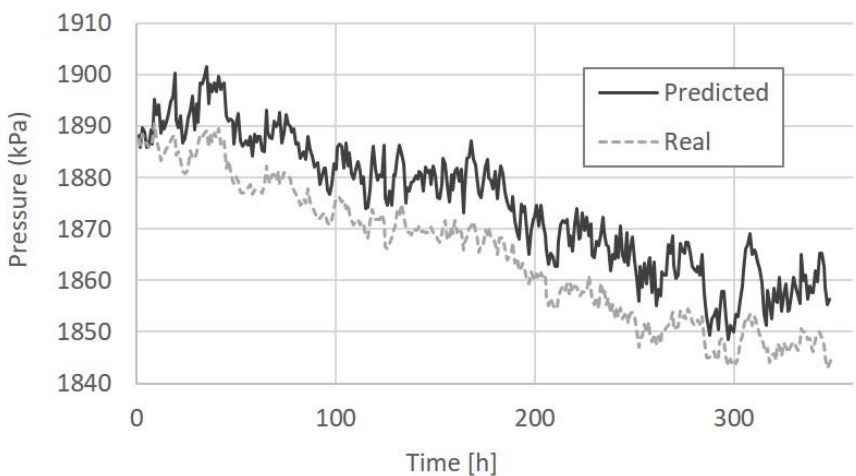

**Figure 17.** Predicted value of P3 vs. actual value.

**Table 4.** Diagnostics system accuracy for fleet data.

| Fault Type | True Positive Rate | False Positive Rate | True Negative Rate | False Negative Rate |
|---|---|---|---|---|
| CF | 99% | 4% | 96% | 1% |
| TE | 99% | 3.5% | 96.5% | 1% |
| BV | 99% | 0% | 100% | 1% |

## 4. Conclusions

A multi-layer, hierarchical approach was proposed in this work to detect and identify simultaneous gas path components degradation and sudden faults in sensors and components. The approach was based on Bayesian network (BN) models, both static and dynamic, and the multi-layer feature allowed to limit the size of conditional probability tables (CPTs) in each BN and still monitor multiple faults simultaneously. The first two layers of the diagnostics system were composed of a physics-based, adaptive performance model to correct the data with respect to ambient conditions, operating conditions, and baseline degradation conditions. Gradual degradation and rapid faults were discriminated at the BN layers based on the fact that the probability of a sharp performance deviation due to gradual wear or material deposition was considered almost null. Instead, abrupt performance deviations compared to the previous condition (be healthy or degraded) are to be attributed to an abrupt fault such as a valve leakage or a sensor malfunction, among others.

The proposed system was tested on synthetic and field data, on three scenarios that comprised of compressor fouling, turbine erosion, bleed valve (BV) leakage, and two sensor faults. It is important to note that the test was in this way performed on two gas turbines (a simulated one and a real one) different from the ones used to train the BN models. While the results from the performance model layers were used to detect a change in performance, the BNs were used to isolate and identify the correct fault. The faults were correctly isolated all times and their magnitude identified mostly correctly, with some small errors due to e.g., presence of noise or model uncertainty. For example, BV leakage was identified with a lower magnitude for 25% of the time in the simulated scenario. Nevertheless, the application on real field data showed the potential of this

approach to aid the plant operators and service staff. The adaptability to different machines in the fleet was also proven. Overall, the maximum error in identification of the fault severity was 1% (i.e., a severity just below or just above the correct one was estimated), but the true positive rate in detection and isolation was around 99% for all cases. A false positive rate below 5% can be still considered high and needs to be improved. However, it concerns estimation of low degradation severity, which does not present any requirements for immediate maintenance actions.

Further, the whole approach was proven to be fairly robust to sensor failures, since the missing measurement could be estimated by the model, leaving some time to perform the necessary maintenance actions. Future work needs to assess the impact of the diagnostics errors and uncertainty on the prediction and planning of future operations, to better understand the limitations of the presented approach.

**Author Contributions:** Conceptualization, V.Z., A.D.F. and K.K.; methodology, V.Z.; formal analysis, V.Z. and A.D.F.; writing—original draft preparation, V.Z.; writing—review and editing, A.D.F. and K.K.; project administration, K.K. All authors have read and agreed to the published version of the manuscript.

**Funding:** This research was funded by the Swedish Knowledge Foundation (KKS) under the project PROGNOSIS, grant number 20190994.

**Institutional Review Board Statement:** Not applicable.

**Informed Consent Statement:** Not applicable.

**Data Availability Statement:** Not applicable.

**Acknowledgments:** The authors would like to acknowledge Anna Sjunnesson and Andreas Hansson from Siemens Energy for their support and for providing the data used in this paper. The authors would also like to thank Mikael Stenfelt and Xin Zhao for their support with the model development.

**Conflicts of Interest:** The authors declare no conflict of interest. The funders had no role in the design of the study; in the collection, analyses, or interpretation of data; in the writing of the manuscript, or in the decision to publish the results.

## Nomenclature

Acronyms
| | |
|---|---|
| ANN | Artificial neural network |
| BN | Bayesian network |
| BV | Bleed valve |
| CF | Compressor fouling |
| CPT | Conditional probability table |
| DBN | Dynamic Bayesian network |
| GPA | Gas path analysis |
| GT | Gas turbine |
| H | High |
| IGV | Inlet guide vane |
| L | Low |
| M | Medium |
| N | Normal |
| TE | Turbine erosion |
| VH | Very high |
| VL | Very low |

Symbols and Greek Letter

| | |
|---|---|
| $r$ | Residual |
| $S$ | Fault severity |
| $\overline{w}$ | Flow capacity |
| $z$ | Measurement |
| η | Efficiency |
| Subscripts | |
| ref | Reference conditions |
| $t$ | Time |

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
