# Peer review of "Assessment of Dynamic Bayesian Models for Gas Turbine Diagnostics, Part 2: Discrimination of Gradual Degradation and Rapid Faults"

_machines, doi:10.3390/machines9120308_

Round 1

Reviewer 1 Report

Thank you for submitting an interesting manuscript for review.

The manuscript discusses the Bayesian networks use for detection of gradual and abrupt faults in an industrial gas turbine. It has potential to be published in Machines. However, several improvements are needed to be implemented before publication of the work. 

1. The authors' choice to divide the article in 2 parts in unusual. Not sure if the second part should be published after the first is approved. 

2. Line 43 "mechanical wear and tear..." - I guess authors thought of fatigue crack, not a tear, that in gas turbine is usually a result of sudden damage, such as foreign or domestic object impact. Please clarify. 

3. Lines 44-46. Is sensor malfunction an actual fault or just a false positive fault alarm? I'd be cautious to put the real, physical damages and sensor malfunctions in one box for automatic sygnal detection. We would like to distinguish easy to fix sensor malfunctions from extensive turbomachinery damage. 

4. There is no explenation, why the specific 4 degradation modes were chosen (compressor fouling, turbine erosion, bleed valve leakage and sensors faults). Please clarify the reason and explain choice. 

5. In test scenarios are there only field data from 1 gas turbine? This does not seem to be representative. Please state clearly why only limitted amount of data was chosen and how it impacts results. 

6. Lines 205-212. Was the gradual degradation included in the scenarios 2 and 3 or just an abrupt failure? Please clarify. 

7. Figure 4 - lack of [%] mark on one of the vertical axes. 

8. Line 231. 5000h? Not 5700h? 

9. Figure 7 and 8. No explenation for N, VL, L, M, delta_eta_c, etc. on vertical axes. 

10. There is no explenation why compressor efficiency rises on Figure 9. Please clarify. 

11. The quality of the figures is low. They look like print screens from MS Excel spreadsheet. There is a number of engineering data visualisation tools that guarantee superior quality. 

12. Figure 12, 13 do not provide descriptions for vertical axes. 

13. No reference provided in the Conclusions, suggesting the results are supported by any other research. 

14. The novelty of the manuscript should be described in more detail.

15. The "Conclusions" section should be revised in the manuscript summarizing the results from the 3 researched scenarios with both qualitative and quantitative approach and providing an outlook on future research.

Author Response

Thank you very much for the valuable comments. We have tried to address them and we hope the changes (marked in red in the manuscript) are satisfactory. Our detailed response is in the attached file.

Reviewer 2 Report

The authors present a hybrid diagnostic method to predict the performance degradation of gas turbine engines. The topic is of interest to the gas turbine community and well aligned with the scope and aim of the journal. The paper is well written and presented.

The use of adaptive model and Bayesian networks constitute a powerful combination for detecting gas turbine faults.

The method has been applied to a real service engine with very good results.

Detailed comments

  1. Page 3 Line 105. The authors state that the compressor air mass flow rate W2 is considered a measurement that is obtained through a bellmouth. In practice, air mass flow rate is not always available as a measurement and can only by obtained through calculation. This is the case for the majority of published works on gas turbine diagnostics.

  1. It would be useful for the reader if the authors could comment on the following:
  • At which extent would the exclusion of this measurement impact the accuracy of the diagnosis?
  • What other measurement could be used to compensate for the above?

Author Response

Thank you very much for the comment, this is a very valid point. A paragraph was added in the text (marked in red in the manuscript) to address this point:

“It is worth to note that W2 is not always measured in gas turbines, requiring a modification of the matching scheme proposed here. Given the high correlation between W2 and the compressor flow capacity change, this deviation may be more difficult to identify correctly with a reduced set of sensors. For two-shaft machines, the rotational speed of the high-pressure turbine could be used instead.”

The selection of optimal measurements for diagnostics was performed by following a method similar to ref [12]. This aspect was also clarified in the text. Therefore, any deviation from the optimal set of measurement would indeed compromise the diagnostics accuracy. However, to some extent, the BN can be trained to detect faults also from a limited number of sensors, as long as the available sensors carry some “unique signature” for each fault. 

Reviewer 3 Report

Following part I of the paper, this part of the manuscript is to integrate the two diagnostic methods to form a multi-layer system. The presented case studies confirm the effectiveness of the integration approach. However, the following minor rooms need to be addressed by the authors before the paper could be considered for publication in the Machines:

1- Although it is generally OK, bear in mind that each part of the paper should be an independent, standalone piece of work. So, some brief descriptions could be added to this part of the paper (from part I) to enhance the independence of this part of the manuscript.

2- There are some typos and grammatical errors in the manuscript. So, the authors are recommended to go through their work once again from the English language point of view. 

Author Response

Thank you very much for the valuable comments. We have tried to address them in the revised manuscript. Changes are marked in red for better visibility, and here a more detailed response to the comments follows:

1- Although it is generally OK, bear in mind that each part of the paper should be an independent, standalone piece of work. So, some brief descriptions could be added to this part of the paper (from part I) to enhance the independence of this part of the manuscript.

Thank you for the suggestion, this issue was addressed by including missing information, without anyway causing too much overlap between the two parts.

2- There are some typos and grammatical errors in the manuscript. So, the authors are recommended to go through their work once again from the English language point of view.

We have carefully checked the language and corrected some minor typos and grammar mistakes. Thank you for pointing this out.